# Virus Inactivation by Formaldehyde and Common Lysis Buffers

**DOI:** 10.3390/v15081693

**Published:** 2023-08-04

**Authors:** Ulrike Seeburg, Lorena Urda, Fabian Otte, Martin J. Lett, Silvia Caimi, Christian Mittelholzer, Thomas Klimkait

**Affiliations:** Molecular Virology, Department Biomedicine, University of Basel, 4009 Basel, Switzerland; ulrike.seeburg@web.de (U.S.); lorena.urda@unibas.ch (L.U.); fabian.otte@unibas.ch (F.O.); martin.lett@unibas.ch (M.J.L.); silvia.caimi@bluewin.ch (S.C.); christian.mittelholzer@unibas.ch (C.M.)

**Keywords:** virus, biosafety, inactivation

## Abstract

Numerous mammalian viruses are routinely analyzed in clinical diagnostic laboratories around the globe or serve as indispensable model systems in viral research. Potentially infectious viral entities are handled as blood, biopsies, or cell and tissue culture samples. Countless protocols describe methods for virus fixation and inactivation, yet for many, a formal proof of safety and completeness of inactivation remains to be shown. While modern nucleic acid extraction methods work quite effectively, data are largely lacking on possible residual viral infectivity, e.g., when assessed after extended culture times, which maximizes the sensitivity for low levels of residual infectiousness. Therefore, we examined the potency and completeness of inactivation procedures on virus-containing specimens when applying commonly used fixatives like formaldehyde or nucleic acid extraction/lysis buffers. Typical representatives of different virus classes, including RNA and DNA viruses, enveloped and non-enveloped, such as adenovirus, enterovirus, lentivirus, and coronavirus, were used, and the reduction in the in vitro infectiousness was assessed for standard protocols. Overall, a 30-minute incubation with formaldehyde at room temperature effectively inactivated all tested enveloped and non-enveloped viruses. Full inactivation of HIV-1 and ECHO-11 was also achieved with all buffers in the test, whereas for SARS-CoV-2 and AdV-5, only five of the seven lysis buffers were fully effective under the tested conditions.

## 1. Introduction

When determining the biological safety of inactivation processes, the quantitative assessment of possible residual viral infectivity is critical. This is of particular importance when specimens containing BSL-3 or -4 level pathogens ([1] for Switzerland as well as other countries with similar biosafety regulatory requirements) are transferred as “non-infectious samples” to a lower biosafety level. National guidelines regulate fixation or treatment procedures in a high containment location prior to a transfer to another laboratory of a lower safety level. While manufacturing guidelines principally focus on the complete inactivation and removal of viruses [2,3,4], treatment of research and diagnostic samples often use inactivation procedures for viruses that aim at retaining properties such as their shape and genomic intactness for downstream analyses in a BSL-2 or -1 environment. Such downgrading critically depends on the reliability of inactivation protocols. One potential hurdle for a simple extrapolation from one virus type to a more general statement is the great heterogeneity between virus classes (DNA vs. RNA or enveloped versus non-enveloped, etc.). But also differences in viral titers or the nature of a specimen from a given diagnostic or research context are likely to lead to varying inactivation success.

This study therefore aimed at providing reliable information on the completeness of given inactivation processes for virus-containing specimens that can inform about potential residual risks of infectiousness of a treated sample.

It is a central fact that viruses can be quite stable in their natural environment. Other groups [5] as well as this study report that even enveloped viruses such as HIV or SARS-CoV-2 can be very stable over time under laboratory conditions [6] and that certain viruses may resist standard chemical detergents for extended periods of time [7,8]. This topic has been extensively discussed, e.g., in a working group of the European virus archiving consortium ‘EVA-GLOBAL (www.european-virus-archive.com (accessed on 25 July 2023)). Therefore, a careful evaluation of the methods used for eliminating residual infectivity is deemed necessary, but only little systematic information is currently available for the different virus orders and families. Hence, validation of common lysis buffers for relevant viruses in diverse specimens including protein-containing liquids is needed, which confirms buffer potencies and provides information about possible residual risks of sample infectiousness.

In this study, we simulated standard conditions of laboratory samples and clinical specimens and compared seven common lysis buffers as well as formaldehyde for their respective potencies to inactivate viruses, using representatives of different virus families including enveloped and non-enveloped viruses, such as lentivirus, coronavirus, adenovirus, and enterovirus.

With respect to specific denaturing agents, guanidinium thiocyanate (C_2_H_6_N_4_S) or ammonium thiocyanate (NH_4_SCN), contained in lysis buffers, are chaotropic agents and therefore act as general protein denaturants that disrupt the hydrogen bonding network between water molecules and reduce the stability of the native state of proteins by weakening the hydrophobic effect. In buffer MC136A, 1-thioglycerol is present as a reducing agent. Formaldehyde (HCHO) is a potent electrophile that can react with biological nucleophiles in proteins and DNA and facilitates the formation of intra-strand and DNA–protein cross-links in vitro [9].

## 2. Materials and Methods

### 2.1. Formaldehyde Inactivation Solution

The formaldehyde (FA) inactivation solution was prepared from a 37% stock of formaldehyde (Sigma-Aldrich, Buchs, Switzerland, Cat. No. 33220) diluted 1:10 in PBS (Sigma, D8537), leading to the final FA concentration of 3.7% added to the infected cells (without media) in the inactivation protocol.

### 2.2. Lysis Buffers

The lysis buffers for nucleic acid extraction in this test panel included: AVL buffer (Qiagen), RLT buffer (Qiagen), and TRIzol Reagent (Invitrogen), and the four lysis buffers A671, MC136A, MC143A, and MC501C (Promega). Details are listed in Table 1.

### 2.3. Viruses

The panel of commonly used virus candidates contains a non-enveloped DNA virus: adenovirus type 5 (AdV-5, Adenovirus), obtained through E. Moissonnier, University Aix-Marseille, Marseille France; a metabolically very stable picornavirus: enteric cytopathic human orphan virus type 11 (ECHO-11, Enterovirus), provided by C. Tapparelle, HUG, Geneva, Switzerland; the enveloped retrovirus HIV-1 (clonal subtype B isolate NL4-3, Lentivirus), originally from M. Martin, NIAID, Bethesda, MD, USA; and a coronavirus (SARS-CoV-2, Wuhan, isolate Muc), provided by G. Kochs, University of Freiburg, Freiburg, Germany. Key properties and the sources of virus strains are summarized in Table 2.

### 2.4. Cell Culture and Virus Stocks

All cell types were grown in complete DMEM (cDMEM): DMEM (Sigma, D0819) supplemented with 10% fetal bovine serum (FBS), (Gibco, 10270) and 1% penicillin/streptomycin (Bioconcept AG, Allschwil, Switzerland, 4-01F00H), or in complete RPMI medium (cPRMI): RPMI 1640 (Sigma) supplemented with 10% FBS and 1% penicillin/streptomycin.

Virus infections were inoculated in the corresponding standard growth media, except for SARS-CoV-2, which utilized DMEM with 2% FBS and 1% penicillin/streptomycin. Cells were incubated at 37 °C with 5% CO_2_. The term “cell culture media” (CCM) refers to both cDMEM or cRPMI. Initial virus inocula were expanded in susceptible cell lines (Table 3) to provide maximal virus titers and stocks. AdV-5 and ECHO-11 were propagated on 90% confluent cultures of suitable cells in T75 flasks. cDMEM was completely aspirated, and cells were infected in 2 mL of cDMEM plus virus at an MOI = 1 in 100 µL for 1 h at 37 °C. Then, 12 mL of cDMEM was re-added to the cultures. When reaching a full CPE (complete cell detachment), virus was harvested on day 3 (AdV-5) or day 6 (ECHO-11). The cell suspension was transferred to a 50 mL tube, subjected to one freeze/thaw cycle on dry ice, and centrifuged at 4 °C, 2000× *g* for 5 min. Aliquots of 1 mL volume were prepared and stored at −80 °C. SARS-CoV-2 was propagated until a full CPE developed. Virus was then harvested from the cell-free supernatant on days 7–12 without freeze–thawing and centrifuged at 1200× *g*, 4 °C for 10 min. HIV-1 stocks were prepared from the chronically infected clonal HUT4-3 cell line [10] in complete RPMI medium on day 3 after seeding 1.5 × 10^5^ cells/mL in 115 mL in T1175 flasks. The cell suspension was transferred to a 50 mL tube and centrifuged (2000× *g*, 5 min). Virus-containing supernatant was aliquoted in 2 mL tubes, which were then high-speed centrifuged for 1 h at 4 °C at 21,100× *g* for virus concentration. After aspirating the top 90% volume from each centrifuged tube, the remaining 10% volume was collected; 1 mL aliquots were prepared and frozen (−80 °C). The propagated virus stocks were used for titer determination by plaque assay (AdV-5, ECHO-11; SARS-CoV-2) or a virus-specific reporter assay (ONPG-conversion for HIV-1, [11]).

### 2.5. Interfering Substances

In the laboratory setting, viral infections are mostly conducted in defined culture media. However, the composition can be quite different when clinical specimens or samples from animal experimentation are handled. Therefore, we applied also conditions with various degrees of “dirtiness” that attempt to mimic the potentially interfering substances anticipated for such samples. With five distinct conditions, differences in viral inactivation with increasing levels of protein and other interfering components (Table 4) were compared.

Stock solutions of all interfering substances were produced as 2× concentrates. BSA (Sigma) was dissolved in DMEM (without the addition of FBS or Pen/Strep), and the solution was supplemented according to the level of dirtiness with sheep erythrocytes (Fiebig Naehrstofftechnik, Germany) or yeast extract (Sigma). Solutions were sterile-filtered (0.22 µm; Sarstedt) and kept in the refrigerator.

### 2.6. Experimental Procedure (Formaldehyde)

The principal inactivation procedure followed four sequential steps: First, cells were infected with the titrated virus inoculum. In the second step, infected cultures were treated with formaldehyde. In the third step, adherent cells were harvested at different time points by scraping. In a fourth step, aiming at the detection of any residual infectivity in the chemically treated samples, aliquots from the cell scrapings were added to growing, sub-confluent cultures. This process was repeated for three consecutive blind cell passages.

In addition, the putative residual virus was serially diluted for blind titration assays. To ensure reproducibility, each inactivation but also the blind passages and virus titrations were executed as three independent experiments, each one in duplicates. For SARS-CoV-2, direct replicates were used.

The formaldehyde inactivation was performed only on cells using a high MOI of the respective virus to test the activity limits of this substance. In our experience, the treatment of cell-free virus supernatant is at least as efficient as the inactivation of the corresponding infected cells. A separate experiment was therefore not included. Incubation times and concentration ranges were chosen as described in common laboratory protocols.

#### 2.6.1. Cell Culture for Titration and Blind Passages

On the day before infection, three 12-well plates were equally seeded with the cell line of interest. In addition, one 96-well plate was seeded for virus titration. The applied seeding densities of the respective cell types are listed in Table 5.

#### 2.6.2. Infection

The cell culture medium was completely removed from the 12-well plates. Cells were then infected at multiplicities of infection (MOI) of 100 (AdV-5), 300 (ECHO-11), 0.2 (HIV-1) or 2.0 (SARS-CoV-2) in final volumes of 200–300 µL of culture media per well and incubated for 1 h at 37 °C, 5% CO_2_. Then, 0.5–2 mL CCM were added per well and incubated for another hour in the same conditions.

#### 2.6.3. Experimental Controls

For solidly verifying a lysis-buffer-driven viral inactivation, various positive and negative controls were included in the same experiments along with the matter of interest, the “treated sample (TC)”:

“Stock control (Stock)”: A total of 20 µL of crude virus inoculum was used directly in the titration. This control served as “input-reference” for any titer drop in the process, to be compared with the treated samples.

“Infected cells without treatment (CCM+)”: Used as mock inactivation, cells were infected and treated with cell culture medium instead of formaldehyde.

“Water control (H_2_O+)”: Infected cells were lysed only with deionized water. This treatment was included as a control for the input virus content in the culture (liberated as well as intracellular particles). For harvesting virus from this control, the medium was completely removed from the well prior to adding 1 mL of sterile milliQ water. After 20 min of incubation to allow the complete osmotic lysis of the cells, the entire content of the well was collected by repeated up-and-down pipetting and transferred to a 1.5 mL tube for titration. Of note, virus losses that could have occurred during the incubation period were not estimated, and no inhibitors to prevent virus degradation by cellular enzymes were added.

“Cell toxicity control (Ctox)”: To assess the impact of any remaining traces of formaldehyde after the cleaning step, uninfected cells were incubated with cell culture medium instead of virus and then treated with formaldehyde for 30 min.

“Negative control (CCM-)”: Uninfected cells were treated with complete medium instead of formaldehyde for the indicated 5 to 30 min. This provided information about the overall cell viability of the cultures.

As “positive control (PC)”, we used a mixture of cell-free virus with interfering substances, yet without any further treatment.

The “T0 (no lysis) control” was manipulated the same way as the treated samples but using water instead of lysis buffer and without the addition of ethanol.

The “TD control” was similar to T0 but did not undergo a centrifugation step in the removal column. This control is intended to observe any virus loss by the column passage.

The “negative control (IS NC)” determined a possible impact of the interfering substances on the viability of the uninfected cells.

To exclude any possible influence of the buffers used, all control samples were diluted exactly the same way as the treated samples.

#### 2.6.4. Inactivation and Cell Collection

For the treatment procedure, the entire culture supernatant was removed from the cultures in the 12-well plate. Then, 1 mL of formaldehyde 3.7% (FA) or the respective control solutions was added to the corresponding wells.

The incubation times for viral inactivation were either 1, 15, 30, or 60 min (AdV-5, HIV) or 15, 30, 60, and 120 min (ECHO-11) since a higher stability of the latter virus was anticipated. SARS-CoV-2 was incubated for 10, 15, 30, or 60 min. One additional well was treated for 5 min with formaldehyde (5’FA) for each virus.

The treatment solutions FA or CCM were removed at the end of the incubation, and cells were gently washed three times with 1 mL PBS to minimize toxicity from residual formaldehyde. PBS was completely removed after the last washing step. Then, 1 mL CCM was added to the wells and cells were scraped off. The entire content of each well was collected in a 1.5 mL tube for further use in titration experiments or blind passages.

#### 2.6.5. Virus Titration

Cells were seeded one day prior to titration in a 96-well plate as stated in Table 5. The titrations included the following conditions: virus stock (stock), water control (H_2_O+), infected cells without treatment (CCM+), and 5 min of formaldehyde treatment (5′FA).

For AdV-5 and ECHO-11, the titration started with an undiluted sample in the first row followed by a serial dilution in log steps (10^−1^ to 10^−10^). For SARS-CoV-2, the titration started with a 10^−1^ dilution in the first row followed by a serial dilution in half-log steps. The last column served as a negative control without infection or treatment.

For HIV-1, a spinoculation step was introduced to obtain higher infection rates, in which the viral particles are forced onto the cell layer, by centrifuging the plates for 90 min at 800× *g* and 25 °C.

Infections were cultivated for 7 days after infection: for AdV-5 or ECHO-11, in final volumes of 200 µL/well, and for SARS-CoV-2, in 110 µL. HIV-1 infected cultures were incubated for 10 days at 37 °C, 5% CO_2_ in a total volume of 220 µL/well.

For the determination of viral loads in the HIV-1 stability experiments (Figure 1), viral reverse transcriptase activity was quantitatively determined as described earlier [12].

HIV titers were determined by a virus-specific reporter assay (ONPG-conversion for HIV-1). Titers of SARS-CoV-2 (TCID_50_/mL) were determined on day 7 as described in the evaluation section using the Spearman–Karber formula [13].
TCID_50_ = 10^X_0_ − d/2 + (d × (sum r_i_)/n_i_)^
X_0_ = positive logarithm of the highest dilution at which all wells are positive
D = dosis distance in log
n_i_ = number of repeats per dilution
r_i_ = the sum of all positive wells starting from X_0_

#### 2.6.6. Blind Passaging

Cells were seeded one day prior to the start of the blind passages as stated in Table 5. The blind passages included the following conditions: different timepoints of FA treatment and a positive control without treatment (CCM+) on a separate plate.

By microscopic inspection, the development of a cytopathic effect (CPE) was monitored. At day 7, 1 mL of supernatant was transferred to another culture plate with uninfected cells (passage 2) and similarly for a third passage. These successive 7-day passages were performed at 37 °C, 5% CO_2_ with the same procedure for all viruses on their respective host cell as indicated.

### 2.7. Experimental Procedure (Lysis Buffers)

#### 2.7.1. Virus Inactivation

For evaluating the buffers AVL, RLT, and TRIzol, the following viruses were used: AdV-5, ECHO-11, HIV-1 as well as SARS-CoV-2. For buffers A671, MC136A, MC143A and MC501C, the viruses AdV-5, HIV-1, and SARS-CoV-2 were tested.

The general inactivation protocol contained four handling steps, which were followed by a stepwise titration for detecting residual infectivity of the viral extracts and by three blind passages to recover residual low titer virus.

In the first step, the virus was mixed with the potentially interfering substance stock solution in equal amounts to create the desired dirty conditions, which were designed to represent specimens with different protein content. In the second step, the respective lysis buffer was added, which was followed by the indicated incubation time. Additional chemicals (e.g., ethanol, chloroform) were added only where required, but the suggested heat or proteinase-K treatment was omitted for buffers MC136A, MC143A, and MC501C. In a third step, the sample was added to an Amicon Ultra-4 column, 100 kDa (Merck AG, Zug, Switzerland) in a final volume of 4 mL, inverted 3 times, and centrifuged (3200× *g*, 15 min) to remove residual traces of lysis buffer and chemicals. Viruses were then recovered from the membrane of the removal column by aspiration as per the manufacturer’s instructions after washing the membrane with 1 mL CCM. The virus was diluted to the intended concentration and plated on susceptible cells (to avoid cytotoxicity, these dilutions had been ascertained in a first round of infections). For buffers A671, MC143A, and MC501C, further dilutions were found to be necessary to eliminate residual toxicity: two dilutions (1:10, 1:200) were used for the blind passages to determine the remaining viral infectivity.

#### 2.7.2. Lysis Conditions

AVL—To 100 µL virus, mixed with interfering substance, 400 µL AVL buffer was added and incubated for 10 min; then, 400 µL ethanol, 96% (Roth AG, Arlesheim, Switzerland), was added. The treated sample was transferred to an Amicon removal column as described above. The procedure was repeated with a second column by transferring the 1 mL extract from the first column to a second column and repeating the steps as described above. The recovered column extract was diluted 1:5.

RLT—For sample treatment, the culture supernatant was removed from one well in the 24-well plate, and 700 µL RLT buffer was added to the cells. The cells were then scraped off and transferred to a 1.5 mL screw-cap tube. The suspension was vortexed for 1 min. Then, 700 µL ethanol (70%) was added to the tube and centrifuged (300× *g*, 1 min) to pellet cell debris. The cell-free supernatant was run through an Amicon removal column as described above, and the recovered column extract was diluted 1:10.

MC136A: To 500 µL virus mixed with interfering substance, 230 µL MC136A buffer was added; then, the tube was vortexed for 5 s and incubated for 15 min (37 °C). The treated sample was run through two consecutive Amicon removal columns as described above, and the recovered column extract was diluted 1:10.

MC501C: To 300 µL virus mixed with interfering substance, 330 µL MC501C buffer was added; then, the tube was vortexed for 10 s and incubated for 10 min. The treated sample was run through two consecutive Amicon removal columns as described above and the recovered column extract was diluted 1:10 (and 1:200).

Before applying A671, MC143A, or RLT to the infected cultures as per protocol, cells were infected with the respective virus inoculum. In order to infect the cells, they were trypsinized (Bioconcept), pelleted, resuspended in 5 mL PBS (Sigma), and counted. The suspensions were divided into two 50 mL tubes with 5 × 10^6^ cells each, pelleted again, resuspended in the corresponding volume of virus for the required MOI or the same volume of cDMEM (uninfected control), respectively, and incubated for one hour (37 °C). Afterwards, the cells were pelleted, washed in 9 mL cDMEM, pelleted again, and resuspended in 12.5 mL cDMEM. The cell suspensions were seeded into two 24-well plates (one plate infected cells, one plate uninfected cells) with 0.5 mL/well. The plates were incubated for 24 h. Mock-treated uninfected cells served as negative control (NC).

For TRIzol lysis, 100 µL of virus was mixed with interfering substance prior to adding 300 µL TRIzol reagent and incubating for 5 min; 60 µL chloroform (Merck) was added to the sample, and the tube was centrifuged (12,000× *g*, 2 min) for phase separation. The upper and interphase were taken off, run through an Amicon removal column as described for AVL, and the recovered column extract was diluted 1:5.

Before treatment with A671, infected cells from one well of a 24-well plate were scraped off and transferred to a 1.5 mL screw-cap tube. Then, 600 µL A671 buffer was added to the cell suspension. The suspension was mixed, and the tube was centrifuged (300× *g*, 1 min) to pellet cell debris. The cell-free supernatant was run through two consecutive Amicon removal columns as described above. The recovered extract was diluted 1:10 and 1:200.

For MC143A lysis, a volume of 200 µL MC143A buffer was added to the infected cell suspensions, and the tube was briefly vortexed and incubated for 4 min at room temperature. Then, 300 µL of the second buffer component (A826D) was added, the tube was vortexed for 5 s, and after, it was centrifuged (300× *g*, 1 min) to pellet cell debris. The cell-free supernatant was run through two consecutive Amicon removal columns as described above, and the recovered column extract was diluted 1:10 and 1:200.

#### 2.7.3. Virus Titration

All virus samples (TC, T0, TD, PC) were titrated in triplicates. Negative controls (IS NC, cDMEM) were included in quadruplicates. Cultures were trypsinized, and cells were counted and seeded in 96-well plates (Table 5) with overnight incubation. Once the respective inactivation protocol was applied, the treated samples and controls could be processed in non-cytotoxic dilutions. Dilutions were made in 96-well plates: the culture media of the 8 wells of column A of the plate were completely removed. For AdV-5 and ECHO-11, 200 µL of the undiluted virus inoculum was added per well of column A. Then, starting from the wells of column A, 20 µL was transferred to the 180 µL of cDMEM pre-dispensed in the wells of column B. After gently mixing by up-and-down pipetting and the repeated transfer of 20 µL of virus-containing media to the next column of the plate, serial 1:10 dilutions covered the range from undiluted virus to a dilution of 10^−10^ in column G. Column H of each 96-well plate served as an uninfected control: the cell culture media was completely removed, and 200 µL of either control (IS NC, cDMEM) was added per well.

To optimize the infections by HIV-1, the respective plates were centrifuged at 800× *g* for 90 min after adding the virus inoculum (“spinoculation”).

For SARS-CoV-2, the titration started with a 10^−1^ dilution in column A of the plates, which was followed by a serial dilution in half-log steps.

To ensure a complete cellular infection, the plates were incubated for 10 days for HIV-1 or 7 days for all other viruses.

#### 2.7.4. Blind Passaging

As a functional proof of the effective inactivation of all infectious virus after treatment, sequential passaging of treated (infected) cell suspensions/extracts on susceptible uninfected target cells has been established as a mandatory standard procedure. To this end, three sequential passages of treated virus on susceptible cells were performed.

Cells were trypsinized, counted, and seeded in a 6-well plate the previous day in a volume of 2 mL per well. Then, 1 mL of the treated virus sample (TC) as well as the negative controls (IS NC) was added to one well of the 6-well plate. After the completion of each passage, 1 mL of the supernatant was transferred to another 6-well plate with pre-seeded cells in 2 mL (CCM). For each blind passage, the following conditions were included: extracts from lysis buffer treatment, cell toxicity control (Ctox), and negative control (CCM-). By microscopic inspection, the development of a cytopathic effect (CPE) was monitored. At day 7, 1 mL of supernatant was transferred to another culture plate with uninfected cells (passage 2) and similarly for a third passage. These successive 7-day passages were performed at 37 °C, 5% CO_2_ with the same procedure for all viruses on their respective host cell as indicated.

Infected but untreated control cells (CCM+) were added to uninfected host cells in a separate 6-well plate for 2 to 7 days to verify full inoculum infectivity and host-cell susceptibility.

### 2.8. Readout for Formaldehyde and Lysis Buffer Experiments

For AdV-5, ECHO-11, and SARS-CoV-2, all plates were evaluated by assessing the typical CPE by microscopic inspection: At the end of the incubation period, all titration plates were stained with Crystal Violet (Sigma) for titer evaluation by visualizing viral plaques. Cultures were fixed by adding 80 µL formaldehyde (3.7%; Sigma) per well and incubation for 1 h. Then, formaldehyde was removed, and 50 µL 0.5% Crystal Violet solution (Sigma, C6158) was added per well and incubated for 5 min. The plate was then rinsed with water and evaluated by eye (violet: intact cells; clear: CPE on infected cultures). Residual titers were calculated by applying the Spearman–Karber formula.

The HIV-1-infected plates were examined by microscopically monitoring the formation of syncytia in cultures of HIV-susceptible cell lines (co-culture of adherent SXR5 reporter cells with SupT1 lymphocytes in suspension). In addition, HIV-1 titration plates were evaluated using the in-house established HIV-susceptible LTR-lacZ reporter cell line SXR5 (enzymatic ONPG-to-ONP conversion) [11]. For this, the culture supernatant was removed, and 10 µL Glo Lysis buffer (Promega) was added per well and incubated for 10 min. The chromogenic solution per plate follows: 1 mL buffer H (250 mM sucrose, 20 mM Na monophosphate, pH 7.5), 8 mL buffer Z (10 mM KCl, 1 mM MgSO_4_, 50 mM β-mercaptoethanol, 100 mM NaH_2_PO_4_, pH 7.5) and 2 mL ONPG solution (8 mg/mL in buffer Z) were mixed and 80 µL solution was added to each well. Color conversion in the wells developed within 2 h of incubation and was scanned visually when a clear contrast between the wells with infection (yellow) and those with no HIV infection (transparent) became apparent. At this point, absorbance at 405 nm was quantified using a TECAN reader.

## 3. Results

### 3.1. Virus Stability

Applying commercial lysis buffers or approved chemicals in accordance with manufacturers’ protocols to clinical specimens or research samples naturally implies the complete inactivation of viral pathogens. On the other hand, it may surprise to find the stability even of enveloped viruses such as HIV or SARS-CoV-2 quite high over time under standard laboratory conditions. We analyzed cell-free, HIV-1-containing culture supernatant for particle stability, and we found that at 37 °C, supernatant RT enzyme activity as an indicator for the intactness of the viral particle had a half-life of about 2 days, while it remained stable for far more than 10 days at 4 °C (Figure 1A). Also, freeze–thaw cycles (−70 °C to 37 °C) affected the enzymatic stability of RT only moderately with ca. 7% loss per cycle (50% activity after 7–8 cycles, Figure 1B).

The decline of viral infectivity was assessed, using SARS-CoV-2, after incubation for 3 and 7 days in different conditions (Figure 1C). While in plain standard cell culture medium or PBS buffer, infectious titers diminished rapidly, the presence of 5 or 30% BSA was able to greatly reduce the titer decline for at least 7 days.

Moreover, the stability of SARS-CoV-2 virus particles under cell culture conditions appears to be remarkably high, even in the presence of viable, growing cells. To test the resilience of SARS-CoV-2 during exposure to non-permissive cells, 10^5^ infectious particles were added to a culture of HEK293T cells in complete DMEM at two temperatures of 34 °C and 30 °C. Infectious virus was still obtained until day 13 under either condition. This was quite unexpected, since the activity of cellular proteases and nucleases, e.g., from dying cells, was anticipated to inactivate SARS-CoV-2 particles within a short time. The previously published study by Widera et al. supports the observation that SARS-CoV-2 viruses are stable for several weeks during liquid storage [5].

### 3.2. Summary of Results for Formaldehyde Inactivation

Older inactivation protocols for FA suggest an incubation time of 5 min with a 3.7% FA solution. Therefore, in the first test, summarized in Table 6, the standard reduction was determined for the four virus families of Table 2, and the potency of a 5-minute formaldehyde inactivation was expressed as log-titer reduction in the respective virus.

A reduction by >6 log_10_ was consistently observed for ECHO-11 and AdV-5. For HIV-1 with a lower initial titer of 1.6 × 10^5^ TCID50/mL, and SARS-CoV-2 with a lower input titer of 2.8 × 10^6^ TCID50/mL, the observed full virus reduction can safely be expressed as a titer reduction by >4 log_10_ (Table 6, Figure 2). The limit of detection in Figure 2 ranged from 0.2 log TCID50/mL for HIV-1 to 1.2 log TCID50/mL for ECHO-11.

#### 3.2.1. Formaldehyde Titration

Virus titrations were performed to verify the TCID50 values of each positive control and the drop of titer in the treated condition.

A virus reduction exceeding 6 log_10_ was determined for AdV-5 and related to the titer of the initial stock. After FA treatment, the signal fell below the threshold of detection. For ECHO-11, in two out of three experiments, residual infection events remained detectable for the lowest virus dilution.

For HIV, we observed a complete elimination of viral infectivity, translating in a virus reduction by at least 4 logs to levels below the threshold (HIV-1, Figure 2). A higher reduction could not be demonstrated for HIV-1, as the primary virus stock only had a TCID50 of 5 log_10_. For SARS-CoV-2, a titer reduction of 4.3 log_10_ was reached.

A “stock control” of the starting sample with a known titer was included as a reference value for which a TCID50 had been determined before. Titers of the controls are from here on referred to as “experimental titers”, and they are determined under the same conditions as the investigated samples.

Although a carry-over of cytotoxic substances was not expected since the lysates were thoroughly washed before collection, rare events of cytotoxicity in the first dilutions of the titration were suspected. Consequently, those wells displaying apparent cytotoxic changes were disregarded in the analysis.

One of the HIV-1 repeat experiments was excluded due to the absence of virus progeny in the positive control. When comparing initial titers of AdV-5 and HIV-1 to the titers obtained with CCM+ and H_2_O controls, a small difference was observed. This might be explained by effects of the handling process, i.e., during times of incubation, washing, or time needed for sample collection (Table 6, lines “H_2_O”; “CCM+”).

#### 3.2.2. Formaldehyde Blind Passage

In order to confirm the absence of residual infectivity, three sequential blind passages on susceptible cells were performed. During this experimental validation, all positive controls for entero- and adenovirus had produced a clear CPE within 48 h post-infection as expected. In the cytotoxicity controls, incubating cells with the chemical in the absence of virus, some partial cytotoxicity was observed in two out of six test series. This was also observed for individual wells of treated samples during the first round of blind passaging, suggesting a minimal carry-over of the fixative into the cultures. Nevertheless, cytotoxicity could be distinguished from cytopathic effects caused by the respective viruses. This was confirmed in each subsequent passage: upon the further dilution of the cytotoxic agent, no evidence for cytotoxicity was noted in any of the cultures of blind passages 2 or 3.

The absence of viral replication (-) or cytopathic effect (CPE) was judged by microscopic inspection. A more detailed summary of residual virus in every blind passage is given in Table 7. Summarizing the results after three blind passages (BP) for each of the tested incubation periods with formaldehyde (FA), 30 min of FA inactivation eliminated the infectivity of all virus classes in the test.

For ECHO-11, HIV-1, and SARS-CoV-2, even a 15-minute FA incubation time was sufficient to completely inactivate these viruses. However, for AdV-5, the 15-minute incubation was not able to eliminate infectivity, as in one of the three cultures, a CPE emerged during blind passage 3.

Shorter exposure times to FA are not recommended, since they may not reliably inactivate the virus of interest. For example, the brief exposure (1–3 min) to FA was insufficient for AdV-5 and HIV-1, and a 5-minute incubation was not sufficient to inactivate SARS-CoV-2. For the ECO-11 picornavirus, which has been reported to be rather resistant to FA, exposures shorter than 15 min were not analyzed.

### 3.3. Summary of Results for Lysis Buffer Inactivation

In order to adequately evaluate viral inactivation by the lysis buffers, both titration plates and the blind passages (for residual virus activity) were analyzed for a comparison of TCID50/mL.

#### 3.3.1. Virus Titration after Lysis Buffer Treatment

The titrations demonstrated a drop in the TCID50/mL for treated samples versus untreated samples in the dilution series. This study had initially aimed for a decrease in viral titer of at least 6 log_10_ from T0 (no lysis) to the treated sample TC (lysis). This turned out not to be possible for every virus in the test. While T0 values showed some variability, we were able to obtain a sufficiently high virus titer for all viruses, and a titer reduction below the detection limit (0.9–1.9 log_10_/mL) was reached for all treated samples (Figure 3). A reduction by 6 log_10_ could thus only be demonstrated for ECHO-11. For the other viruses in the test, the maximal possible reduction did not exceed 5 log_10_ (AdV-5), 3 log_10_ (HIV-1) or 5 log_10_ (SARS-CoV-2), respectively (Figure 3). For all viruses (AdV-5, ECHO-11, HIV-1, SARS-CoV-2), we compared the treated samples TC (light gray bars, lysis) with the corresponding untreated sample T0 (dark gray bars; no lysis) regarding the decrease in log TCID50/mL. The TCID50 reduction in log varied for the different viruses and lysis buffers. Nevertheless, for all viruses, the sample treated with lysis buffer dropped below the calculated detection limit (dotted line). No trend for the influence of interfering substances could be observed.

#### 3.3.2. Blind Passaging after Lysis Buffer Treatment

The definitive absence of virus replication is not explicitly proven after a single replication cycle, but it has been established to include up to three sequential blind passages on susceptible cells. Similarly, Widera et al. also report that “lysis buffers do not always reach a complete infectivity loss and therefore it is advised to use a second inactivation method to guarantee a safe inactivation” [5]. Therefore, a second inactivation agent such as ethanol is already included in some of the manufacturers’ protocols, e.g., AVL or RLT buffer. Overall, no cytopathic effect (CPE) that would indicate residual virus replication was detected after passaging the cells for more than three weeks in three consecutive passages (Table 8) of enterovirus or HIV-1. The safe lysis of AdV-5 and no sign of virus recovery after three blind passages could only be demonstrated for buffers A671, MC143A and TRIzol in the presence of at least 3 g/L BSA. In addition, AdV-5 was completely neutralized in two out of three experiments by the treatment with buffers (AVL with at least 3 g/L BSA + Ery) or RLT. Buffers (AVL at 3 g BSA/L or below), MC136A and MC501C (lysis of the viral supernatant only tested in the presence of interfering substances) did not fully inactivate AdV-5. An important note: the manufacturer recommends using the lysis buffers MC136A and MC501C in conjunction with a proteinase K/heat treatment!

Treatment with buffer MC136A (again omitting the recommended proteinase K/heat treatment) was unable to completely inactivate SARS-CoV-2, and for MC143A and MC501C, a CPE for SARS-CoV-2 was noted in one of the three blind passage experiments.

### 3.4. Utility of Salt Removal Columns

Any titer reduction that could be caused by the plain passage through an Amicon removal column was assessed by comparing the two positive controls TD (no lysis, no column) to T0 (no lysis, with column passage) samples. Both control samples were similarly treated with water instead of lysis buffer. The reducing effect of the column turned out to be minimal (Figure 4) but tended to depend on the initial virus concentration: while for AdV-5 and HIV-1 (with titers of 1 × 10^8^ TCID50/mL and 1 × 10^5^ TCID50/mL, respectively), we observed only a minimal loss of viral titers after passage through a removal column, the column-passage of ECHO-11 (titer > 1 × 10^11^ TCID50/mL in experimental titration) caused a reduction of 1–3 log_10_. Although not experimentally verified, it is possible that this was due to a limited binding capacity of the columns. For SARS-CoV-2, we found no reducing effect over the column for most buffers except for MC501C with a reduction of 2.2 log_10_. Potential reasons for these differences were not further investigated, but virus-specific differences in the affinity to the resign cannot be excluded.

As expected, for most viruses, only minimal differences between positive control (PC) and TD samples were observed. The virus-reducing effect of the passage through a removal column was more pronounced for ECHO-11 with a titer of (>1 × 10^11^ TCID50/mL) than for AdV-5, HIV, or SARS-CoV-2 (titers 1 × 10^8^ TCID50/mL, 1 × 10^5^ TCID50/mL and 2.8 × 10^6^ TCID50/mL, respectively).

### 3.5. Potential Influence of Interfering Substances on Viral Inactivation

No significant difference in the inactivating potential that might have been caused by the presence of interfering substances was observed (Figure 4). For AdV-5 and ECHO-11, five conditions of “dirtiness” were applied in the presence of AVL buffer, RLT buffer, and TRIzol. A minimal experimental variability in the positive control samples of each experiment comparing the different dirty conditions was likely due to the sequential conduct on different days. Moreover, no dose-dependent impairment was found when increasing amounts of BSA or other interfering contents (sheep erythrocytes, yeast extract) were added (Figure 4). Two or three main ‘dirtiness’ conditions (0.3 g/L BSA; 10 g/L BSA + yeast extract; 80 g/L BSA) were also applied to the lysis of HIV-1, again demonstrating the absence of any activity-reducing effect of high protein concentrations. Based on these negative findings, dirty conditions were not tested for SARS-CoV-2.

Summarizing all results for the lysis buffers (Table 9), no replicative virus was detectable after three successive blind passages for five of seven lysis buffers, confirming the complete virus inactivation by those buffers.

The two lysis buffers MC136A and MC501C were, in the absence of proteinase K/heat, unable to fully inactivate AdV-5; MC143A with no proteinase K/heat did not completely eliminate SARS-CoV-2. Interfering substances mimicking protein conditions in clinical samples did not affect viral inactivation by any of the buffers.

## 4. Discussion

In recent years and during the course of this study, different viral pathogens became relevant in the laboratory or new ones emerged. This resulted in a certain heterogeneity in the test panels for the various viral pathogens. Beyond this, a comprehensive direct side-by-side comparison of a large array of relevant pathogens was not the main focus of this work. We rather intended to examine in detail the possible impact of a common formaldehyde-based fixation buffer or of commercial lysis buffers that are commonly used in research laboratories on preparations of viruses from representative classes. To better reflect the heterogeneous composition of virus-containing specimens, several typical protein conditions as potentially interfering substances were examined.

A successful inactivation was verified using the sensitive method of three consecutive blind virus passages. As any trace of replication-competent virus in one culture would be amplified during the next blind passage, the absence of infectivity after a 3-week observation may be taken as a safe indicator for complete viral inactivation. This sensitive culture method revealed and confirmed that for adenoviruses and SARS coronaviruses, not every lysis buffer may be suitable.

For a chemical treatment with FA, the results validate the full inactivation of high titer stocks of enveloped (HIV-1, SARS-CoV-2) as well as non-enveloped viruses (AdV-5, ECHO-11) with a 30-minute incubation period at room temperature. As the lipid bilayers of the viral envelopes render these virus particles less stable than the protein-coated capsids of adenoviruses or enteroviruses, this is in full agreement with their complete inactivation within 15 min.

Also, the non-enveloped picornavirus ECHO-11 was effectively inactivated within this time but retained some viability when exposed to FA for only 5 min. For AdV-5, another non-enveloped virus candidate, the period of 15 min was not fully sufficient, as in some cultures, replicative AdV-5 virus could be recovered after passage.

The complete inactivation of the enveloped viruses in the panel is in agreement with the property that lipid bilayers are principally highly sensitive to detergents. Accordingly, HIV-1 and SARS-CoV-2 (enveloped) were readily inactivated by the tested lysis buffers. This was also observed for the non-enveloped Echo-11 virus.

In remarkable contrast, AdV-5 was only incompletely inactivated by AVL, RLT, and Trizol, which was an observation also reported by others [14].

For the buffers MC501, MC143A, and MC136A, the thermal or proteolytic inactivation step of the specimens (proteinase K or 56 °C heat treatment) had deliberately been omitted: as it is known that heat is able to affect and inactivate viruses, this study rather focused on the very properties of the chemical lysis buffer compositions. Consequently, insufficient inactivation cannot be concluded due to not adhering to the manufacturer’s protocol. Nevertheless, without the heat- or proteinase-K step, SARS-CoV-2 and AdV-5 are only incompletely inactivated by buffers MC501, MC143A, or MC136A.

This study shows that viral susceptibility to different lysis buffers can greatly vary among virus orders and families. The fact of residual viral infectiousness being likely or possible when using certain lysis buffers/processes must be recognized and taken into account in quality management processes for laboratory work involving viruses.

## 5. Conclusions

This study demonstrates the relevance of a thorough assessment of the reliable inactivation of viruses and the risk of residual infectiousness in samples in laboratory and clinical settings.We demonstrate that not all buffers or fixatives readily inactivate every virus when applying standard conditions, which indicates that, especially for new pathogens, safe conditions must be verified and validated that may require a prior evaluation of the inactivation system to be used.An interference of high protein contents and other potentially interfering supplements as reported by others [2] for inactivation processes was not confirmed with our test panel.

## Figures and Tables

**Figure 1 viruses-15-01693-f001:**
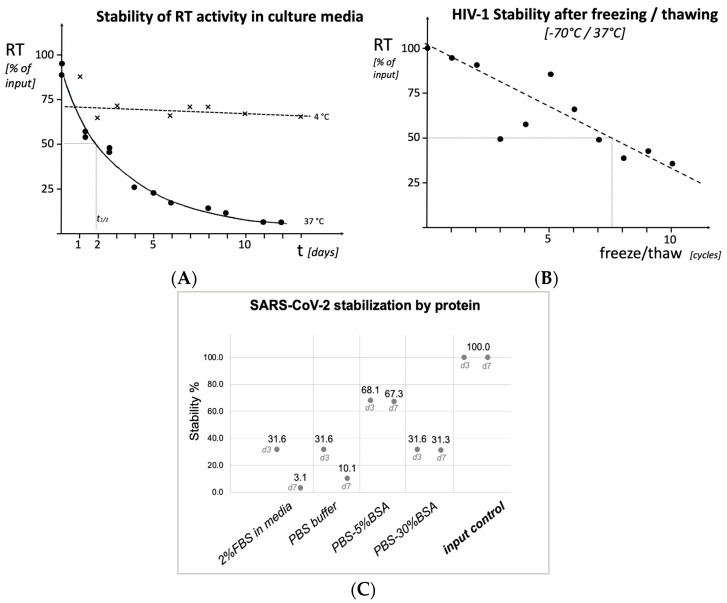
Stability of the HIV Reverse Transcriptase enzyme (HIV-1, (**A**)), of virus particles (HIV-1, (**B**)) and virus infectivity (SARS-CoV-2, (**C**)) after typical laboratory manipulations. (**A**): HIV-1 containing cell-free culture supernatant, kept for 13 days at 4 °C ("x" and dotted line) or at 37 °C (solid symbols, solid line), or (**B**): subjected to repeated cycles of freezing on dry ice, followed by thawing at 37 °C. (**C**): SARS-CoV-2 virus infectivity after 3 or 7 days of incubation as indicated on the *X*-axis.

**Figure 2 viruses-15-01693-f002:**
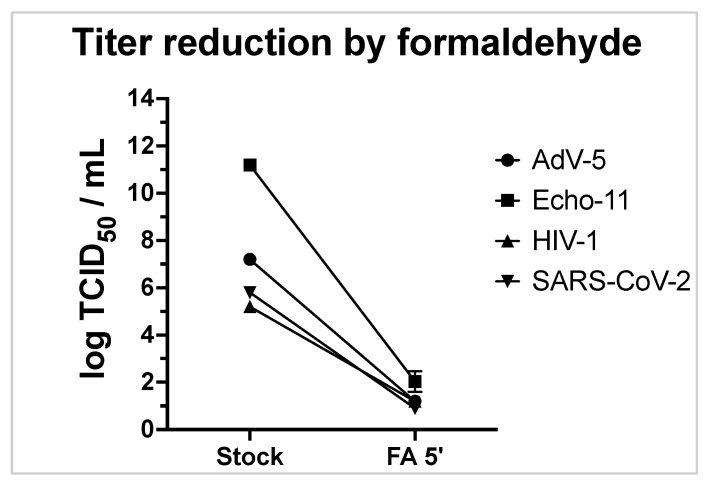
Virus titers (log TCID50/mL) before and after a 5 min exposure to FA is shown for Echo-11 (square symbols), HIV-1 (triangles), AdV-5 (circles), and SARS-CoV-2 (inverted triangle). Independent triplicate experiments (duplicates for SARS-CoV-2) are shown.

**Figure 3 viruses-15-01693-f003:**
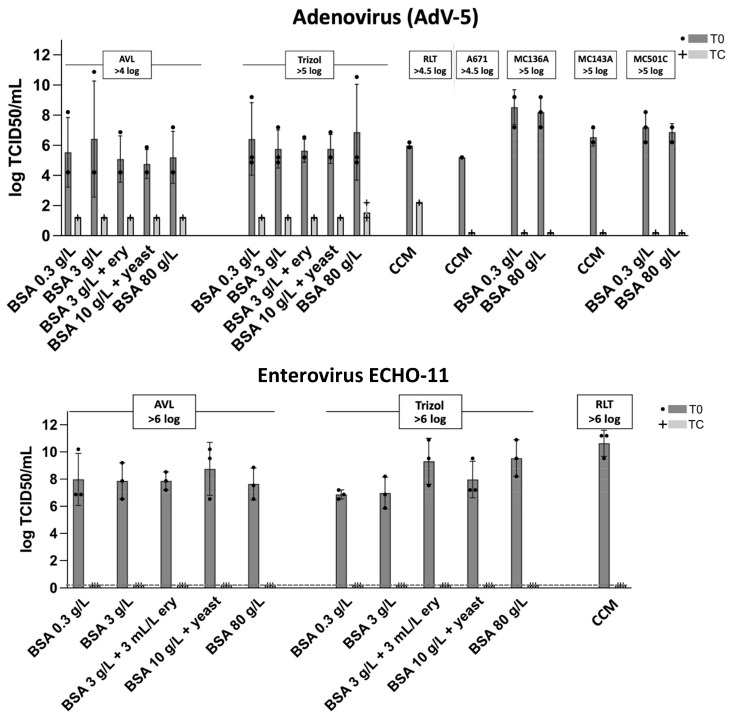
Virus inactivation by lysis buffers. For each of the viruses and conditions, a first aliquot was analyzed prior to adding the test agent (T0, dark gray bars) and a second sample after incubation with the indicated reagent (TC, light gray). For AdV-5, ECHO-11, and HIV-1, also several ‘dirty’ conditions were analyzed as shown. All tests were completed in three independent replicates. Dotted lines indicate the detection limit in the respective assay.

**Figure 4 viruses-15-01693-f004:**
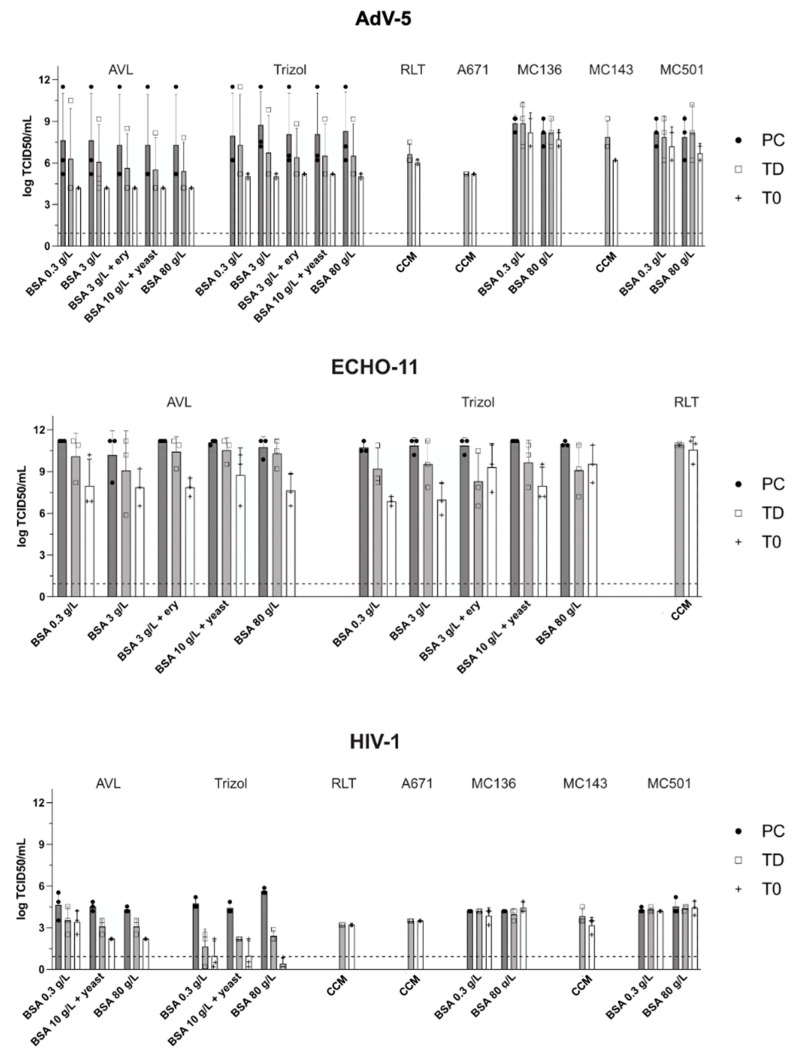
Assessment of the possible influence of the agent-removal column and potential impact of the addition of external protein in ‘dirty’ conditions. “process” indicates, from which procedure the respective samples were taken. Of note, none of the PC, TD or T0 samples had been treated with the indicated reagents. The quantification limit (10^−2^/mL) is indicated by a dotted line. Abbreviations: BSA—bovine serum albumin; “ery”—erythrocyte concentrate added; “CCM”—complete culture media.

**Table 1 viruses-15-01693-t001:** Inactivation buffers.

Name	Company	Extraction Kit/Reference	Active Ingredients	Usage on
AVL	Qiagen	Catalog no. 19073	Guanidinium thiocyanate	viral cell culture supernatant
RLT	Qiagen	Catalog no. 79216	Guanidinium thiocyanate	infected cells
TRIzol	Invitrogen/ Thermo Fisher Scientific	Catalog no. 15596026	Thiocyanic acid, compound with guanidine (1:1) Ammonium thiocyanate	viral cell culture supernatant
A671	Promega	AS1620—Cultured Cells DNA Kit	Guanidinium thiocyanate	infected cells
MC136A	Promega	AS1460—Maxwell® RSC miRNA from Tissue and Plasma/Serum Kit	Guanidine thiocyanate 1-thioglycerol	viral cell culture supernatant
MC143A + A826D	Promega	AS1660—PureFood Pathogen Kit	Guanidinium thiocyanate	infected cells
MC501C	Promega	AS1330—Viral TNA Kit AS1400—Blood Kit	Guanidinium thiocyanate	viral cell culture supernatant

**Table 2 viruses-15-01693-t002:** Human virus strains used in the study.

Genus	Virus	Characteristics/Features	Strain	Biosafety Level
Adenovirus	AdV-5	Non-enveloped, DNA (ds)	-	BSL2
Enterovirus	ECHO-11	Non-enveloped, RNA (ss, +)	-	BSL2
Lentivirus	HIV-1	Enveloped, RNA (ss, +)	NL4-3	BSL3
Coronavirus	SARS-CoV-2	Enveloped, RNA (ss, +)	Wuhan	BSL3

**Table 3 viruses-15-01693-t003:** Propagation and reporter cells.

Virus	Cell Type	Propagation	Readout	Harvest	Final Titer (TCID50)
AdV-5	A549	3 days	day 7	SN + detached cells, freeze/thaw, centrifugation	1 × 10^8^/mL
ECHO-11	Vero	6 days	day 7	SN + detached cells, freeze/thaw, centrifugation	3 × 10^8^/mL
HIV-1	HUT4-3	3 days	-	Remove cells, concentrate virus 10fold by centrifugation	3 × 10^5^/mL
	SXR5&SupT1	-	day 10		
SARS-CoV-2	CaCo2/ VeroE6-T2	12 days/ 4 days	-	SN + detached cells, centrifugation	3 × 10^6^/mL
	VeroE6	-	day 7		

A549: adenocarcinomic human alveolar basal epithelial cells; VeroE6: kidney epithelial cells from African Green Monkey; VeroE6-T2: Vero cells, stably expressing hu-TMPRSS2; HUT4-3: Hut78-derived, stably producing infectious HIV-1; SXR5, HeLa derived human cervical cancer cells; SupT1: human T-cell lymphoblasts; CaCo2: human colon epithelial cells.

**Table 4 viruses-15-01693-t004:** Interfering substances.

Level of “Dirtiness”	Supplement to DMEM
low dirty condition	0.3 g/L BSA
dirty condition	3.0 g/L BSA
dirty condition + erythrocytes	3.0 g/L BSA + 3.0 mL/L sheep erythrocytes
dirty condition + yeast	10.0 g/L BSA + 10.0 g/L yeast extract
high dirty condition	80.0 g/L BSA

**Table 5 viruses-15-01693-t005:** Seeding densities for 12-well plates (12wp) for the respective cell types used for inactivation, titration in 96-well plates (96wp), and for blind passaging in 12wp.

Virus	Cells	Culture Media	12wp Treatment	12wp Blind Passages	96wp Titrations
AdV-5	A549	cDMEM	2 × 10^6^/plate	2 × 10^6^/plate	1 × 10^6^/plate
ECHO-11	Vero	cDMEM	1 × 10^6^/plate	1 × 10^6^/plate	1 × 10^6^/plate
HIV-1	SXR5 SupT1	cDMEM	2.5 × 10^6^/plate	1.5 × 10^6^/plate	8 × 10^5^ plate
		cRPMI	5 × 10^5^/plate	5 × 10^5^/plate	2 × 10^5^/plate
SARS-CoV-2	VeroE6	cDMEM (2% FBS)	2 × 10^6^/plate	2 × 10^6^/plate	1 × 10^6^/plate

**Table 6 viruses-15-01693-t006:** Summary of the “log titer “ for each virus as indicated for each indicated condition: “stock”, water dilution (“H_2_O”), dilution in cell culture media (“CCM+”), or after a 5-minute formaldehyde (FA) treatment. The final virus infectivity (as TCID50/mL) is shown. FA-treated samples boxed in gray; nd = values < 1.2 log omitted due to failing control well. ** indicates that complete inactivation was reached only in one out of three repeat experiments. Titer reduction is shown compared to the initial virus stock.

Virus	Condition	Experiment 1 log TCID50/mL	Experiment 2 log TCID50/mL	Experiment 3 log TCID50/mL	Average TCID50/mL	TiterReduction
AdV-5	Stock	7.2	7.2	7.2	1.58 × 10^7^	
	H_2_O+	4.2	5.2	4.2	1.58 × 10^4^	
	CCM+	4.2	4.2	4.2	1.58 × 10^4^	
	FA 5′	1.2	1.2	1.2	<1.58 × 10^1^	6.0 log_10_
ECHO-11	Stock	11.2	11.2	11.2	1.58 × 10^11^	
	H_2_O+	11.2	11.2	11.2	1.58 × 10^11^	
	CCM+	11.2	10.2	10.2	6.32 × 10^10^	
	FA 5′	2.2	2.7 **	1.2	2.25 × 10^2^	9.2 log_10_ **
HIV-1	Stock	5.2	5.2	5.2	1.58 × 10^5^	
	H_2_O+	nd	2.2	2.2	1.58 × 10^2^	
	CCM+	nd	2.2	4.2	7.98 × 10^3^	
	FA 5′	nd	1.2	1.2	<1.58 × 10^1^	4.0 log_10_
SARS-CoV-2	Stock	5.8	5.8	-	2.8 × 10^6^	
	H_2_O+	4.9	4.9	-	8.1 × 10^4^	
	CCM+	5.4	4.9	-	1.8 × 10^5^	
	FA 5′	0.9	0.9	-	8.1 × 10^0^	4.9 log_10_

**Table 7 viruses-15-01693-t007:** Summary of blind passages in 3 distinct experiments (A–C) after formaldehyde treatment. For each virus, the host cell is given in parentheses. Green boxes indicate the absence of viral replication. Red boxes indicate the formation of a CPE or syncytia (S), which is typical for the respective virus. “-” indicates that the control without FA was stopped after the 1st blind passage.

		------------------A-------------------			------------------B-------------------			-------------------C------------------		
Virus	Condition	BP1	BP2	BP3	BP1	BP2	BP3	BP1	BP2	BP3
AdV-5 (A549)	no FA	CPE	-	-	CPE	-	-	CPE	-	-
	1’ FA		CPE	CPE	CPE	CPE	CPE		CPE	CPE
	15’ FA						CPE			CPE
	30’ FA									
	60’ FA									
ECHO-11 (Vero)	no FA	CPE	-	-	CPE	-	-	CPE	-	-
	15’ FA									
	30’ FA									
	60’ FA		-							
	120’ FA									
HIV-1 (SXR5 & SupT1)	no FA	CPE	-	-	CPE	-	-	CPE	-	-
	1’ FA		S	S		S	S		S	S
	15’ FA									
	30’ FA									
	60’ FA									
SARS-CoV-2 (VeroE6)	no FA	CPE	-	-	CPE	-	-	CPE	-	-
	5’ FA			CPE		CPE	CPE		CPE	CPE
	15’ FA									
	30’ FA									
	60’ FA									

**Table 8 viruses-15-01693-t008:** Summary of blind passages. To test low residual virus in supernatants from treated specimens, these were used as inocula to set up repeated rounds of incubation. This ‘blind supernatant passage’ as a standard method to amplify and reveal low traces of infectious virus is repeated in three successive rounds indicated as p1 to p3 for each buffer. Red boxes are used to indicate a strong virus-typical CPE. Yellow boxes indicate a weak CPE. Green boxes indicate the complete absence of CPE. “-” indicates that no experiment was performed for the respective condition.

	Virus:	AdV-5										HIV-1										ECHO-11										SARS-CoV-2								
Experiment (E):		E1			E2			E3				E1			E2			E3				E1			E2			E3				E1			E2			E3		
Passage Number:		1	2	3	1	2	3	1	2	3		1	2	3	1	2	3	1	2	3		1	2	3	1	2	3	1	2	3		1	2	3	1	2	3	1	2	3
AVL	CCM (clean)	-	-	-	-	-	-	-	-	-		-	-	-	-	-	-	-	-	-		-	-	-	-	-	-	-	-	-										
	0.3 g/L BSA																															-	-	-	-	-	-	-	-	-
	3 g/L BSA											-	-	-	-	-	-	-	-	-												-	-	-	-	-	-	-	-	-
	3 g/L BSA+E											-	-	-	-	-	-	-	-	-												-	-	-	-	-	-	-	-	-
	10 g/L BSA+Y																															-	-	-	-	-	-	-	-	-
	80 g/L BSA																															-	-	-	-	-	-	-	-	-
RLT	CCM (clean)																																							
TRIzol	CCM (clean)	-	-	-	-	-	-	-	-	-		-	-	-	-	-	-	-	-	-		-	-	-	-	-	-	-	-	-										
	0.3 g/L BSA																															-	-	-	-	-	-	-	-	-
	3 g/L BSA											-	-	-	-	-	-	-	-	-												-	-	-	-	-	-	-	-	-
	3 g/L BSA+E											-	-	-	-	-	-	-	-	-												-	-	-	-	-	-	-	-	-
	10 g/L BSA+Y																															-	-	-	-	-	-	-	-	-
	80 g/L BSA																															-	-	-	-	-	-	-	-	-
A671	CCM (clean)																					-	-	-	-	-	-	-	-	-										
MC136A	CCM (clean)	-	-	-	-	-	-	-	-	-		-	-	-	-	-	-	-	-	-		-	-	-	-	-	-	-	-	-										
	0.3 g/L BSA																					-	-	-	-	-	-	-	-	-		-	-	-	-	-	-	-	-	-
	80 g/L BSA																					-	-	-	-	-	-	-	-	-		-	-	-	-	-	-	-	-	-
MC143A	CCM (clean)																					-	-	-	-	-	-	-	-	-										
MC501C	CCM (clean)	-	-	-	-	-	-	-	-	-		-	-	-	-	-	-	-	-	-		-	-	-	-	-	-	-	-	-										
	0.3 g/L BSA																					-	-	-	-	-	-	-	-	-		-	-	-	-	-	-	-	-	-
	80 g/L BSA																					-	-	-	-	-	-	-	-	-		-	-	-	-	-	-	-	-	-

**Table 9 viruses-15-01693-t009:** Virus reduction by various lysis buffers. Starting titers are indicated for each virus; the “log-reduction” value results from the final virus dilution in the culture. The strength of CPE is given as “-”: absent; “+” mild CPE; “++” significant CPE; “+++” full CPE. “?” indicates an indeterminate result. * A heat inactivation step as suggested by the supplier was omitted to test the inactivating properties of the buffers themselves without a thermal contribution.

Buffer	Virus	Stock Titer (TCID50/mL)		CPE Results			Reduction from Mock (TD) log_10_
		Stock	Experiment	p1	p2	p3	
AVL	AdV	1 × 10^8^	1 × 10^8^	>5.5	?	?	>5.5
	ECHO-11	3 × 10^8^	4 × 10^11^	>9.9	-	-	>9.9
	HIV-1	3 × 10^5^	1 × 10^5^	>3	-	-	>3
	SARS-CoV-2	2.8 × 10^6^	2.8 × 10^6^	>4.7	-	-	>4.7
RLT	AdV	1 × 10^8^	1 × 10^8^	>5.3	-	-	>5.3
	ECHO-11	3 × 10^8^	4 × 10^11^	>10.6	-	-	>10.6
	HIV-1	3 × 10^5^	1 × 10^5^	>3.9	-	-	>3.9
	SARS-CoV-2	2.8 × 10^6^	2.8 × 10^6^	>5	-	-	>5
TRIzol	AdV	1 × 10^8^	1 × 10^8^	>6.3	-	-	>6.3
	ECHO-11	3 × 10^8^	4 × 10^11^	>9	-	-	>9
	HIV-1	3 × 10^5^	1 × 10^5^	>1.9	-	-	>1.9
	SARS-CoV-2	2.8 × 10^6^	2.8 × 10^6^	>4.3	-	-	>4.3
A671	AdV	1 × 10^8^	1 × 10^8^	>5	-	-	>5
	HIV-1	3 × 10^5^	1 × 10^5^	>3	-	-	>3
	SARS-CoV-2	2.8 × 10^6^	2.8 × 10^6^	>6	-	-	>6
MC136A	AdV	1 × 10^8^	1 × 10^8^	>8.5	+	+++ *	>8.5
	HIV-1	3 × 10^5^	1 × 10^5^	>3.9	-	-	>3.9
	SARS-CoV-2	2.8 × 10^6^	2.8 × 10^6^	>6.1	+	+++ *	>6.1
MC143A	AdV	1 × 10^8^	1 × 10^8^	>7.7	-	-	>7.7
	HIV-1	3 × 10^5^	1 × 10^5^	>3.6	-	-	>3.6
	SARS-CoV-2	2.8 × 10^6^	2.8 × 10^6^	>5.3	++	+++ *	>5.3
MC501C	AdV	1 × 10^8^	1 × 10^8^	>8	+	+++ *	>8
	HIV-1	3 × 10^5^	1 × 10^5^	>4.2	-	-	>4.2
	SARS-CoV-2	2.8 × 10^6^	2.8 × 10^6^	>7	++	+++ *	>7

## Data Availability

All primary data leading to the presented figures and tables are available from the authors’ laboratory. No new sequence data have been generated that require submission to public repositories. All rights on virus stocks remain with the original providers or, if not stated otherwise, with the authors.

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
