# Peer review of "Virus Inactivation by Formaldehyde and Common Lysis Buffers"

_viruses, 2023, doi:10.3390/v15081693_

Round 1
Reviewer 1 Report (Previous Reviewer 1)
The author has revised all the matters, but there are still some minor mistakes. I think the manuscript can be accepted with these mistakes corrected. And some of these mistakes appear more than once in the manuscript, and the entire manuscript should be examined in more detail.
Line 426 Log TCID50/mL should be Log TCID50/mL
Line 472 a cytopathic effect should be cytopathic effect
Line 487 15-minute should be 15 minutes
The font for Table 1 should be changed to Times New Roman
Author Response
We thank the reviewer very much for pointing our remaining typos and errors in the text (and feel slightly embarrassed that we did not see them in the first place...)
We have now carefully re-worked the manuscript and eliminated also ALL mentioned errors:
- "TCID50/mL" throughout the text
- corrections in lines 472 and 487
- All fonts in text and tables have been adjusted.
Moreover, for higher clarity, also several text paragraphs have been modified without changing the content. also several citations have been added where needed.
Reviewer 2 Report (New Reviewer)
Overall comments
Having clear, published data concerning what kinds of treatments effectively render virus in samples non-infectious would be useful. This paper offers some useful data toward that end, although the number of viruses texted and the inactivation strategies used were not large.
Minor
Line 32 “CH”. Does this mean Switzerland? This should just be spelled out. “European countries” does this mean the EU? European Economic Area? What about UK? US, which is likely the biggest entity? Really, if this manuscript aims to provide an information resources applicable it should do that.
Spelling typo on line 363
What does line 462, “On the side of our experimental validation…” mean?
Introduction
The manuscript should include at least a brief description of the physicochemical basis underlying the inactivation by the different agents.
What is known about the stability/lability of the viruses selected as representative for this study?
There is a very large literature on many different methods of inactivating different viruses. This paper does not do an adequate job of reviewing that preceding work or placing this work into that historical context.
Why were the inactivation methods described in this paper selected for study?
Methods
Table 1 should state what is in the different kits, what the active agents are (e.g. guanidinium thiocyanate (can be found in the material safety data sheets), and provide the actual, marketed names for the kits, not just abbreviations. “ref” should be “catalog number”. Safety material should be given for each agent.
Why were the tested agents/buffers selected?
Line 217, something is missing: “HIV titers were determined by XXXX.”
The experimental procedure for the formaldehyde inactivation is unclear. Was this really just applied to cells producing virus? Why not expose isolated virus to varying concentrations of formaldehyde and look at viral titer reduction. For the formaldehyde inactivation experiments it would be much better to see results from multiple concentrations. How/why was the tested concentration and duration chosen?
For the formaldehyde inactivation procedure, the manuscript should be clear to distinguish between the concentration of formaldehyde that is added to the virus preparation and the final formaldehyde concentration in the virus preparation. This is not done, and it is confusing in this manuscript.
In the studies of Table G, the effectiveness of different lengths of exposure to formaldehyde, the effectiveness of inactivation was judged only by observation of CPE, a weak approach. A much more convincing way to do this would be via a PCR/rt-PCR (as appropriate) assay. Table G should have a negative control (no formaldehyde) for each virus.
Results
Data of Fig 1: the manuscript seems to equate stability of RT enzymatic activity (panels A and B) with stability of SARS-CoV-2 viral infectivity (panel C). These are really two very different things.
Conclusions
The paper introduction starts off saying that the purpose was to determine what treatments could render virus-containing samples safe (“This study therefore aimed at providing reliable information on the completeness of given inactivation processes for virus-containing specimens that is able to inform about potential residual risks of infectiousness of a treated sample.”) The conclusion section does not make a clear, conclusive statement addressing this question.
Figures and Tables
Figure 1 should not be split across 2 pages.
The figures and tables should indicate what the limits of detection for viruses are. As is, the figures give the impression that there is some live virus left after some of the treatments, when this not the case. This seems to have been done for Figures 3 and 4, where there is a dotted line at the 1log10 level, but in this figure this line is not defined in Figure 3.
Table G should indicate at the top that these are 3 distinct experiments.
Acceptable
Author Response
We would like to express our gratitude for this thorough and helpful review!
We have carefully analysed the text on the basis of the critique and suggestions and have now modified the respective sections in the manuscript (highlighted in grey). In detail (point-by-point):
OVERALL: it is true that this report does not represent a most comprehensive analysis of all relevant viral pathogens. Instead and as now clearly indicated, we chose typical candidates that reflect RNA- and DNA viruses, enveloped and non-enveloped viruses. And this choice already shows the need for individual re-testing, when it comes e.g. to newly emerging viruses that might "appear similar to known ones". Here the SARS-CoV-2 sample may serve as example.
LINE 32: Thank you; odd abbreviations have all been eliminated and the text was re-phrased to indicate the "globally available guidelines in most countries" (line 33)
Lines 363 and 462 have been corrected.
INTRO
Important point: A general statement on the chaotropic nature of buffers has been added (line 67-73)
On the stability of the viruses in this study, text was rephrased and added: lines 51-55; 444-445; 584-587.
On large body of literature: Agreed, a comprehensive review would be another option, not chosen for this study; we rather chose the pragmatic way or directly assessing the pathogenic viruses in typical conditions used in diagnostic and research labs. This has been rephrased and addressed in lines 659-667.
Why were these buffers chosen? - There was agreement in the working group of the European EVAg consortium that these are VERY commonly used lysis systems. In recent years, the new buffers, provided by Promega, were added and seemed worthwhile. FA is, of course, a very commonly used fixative, but surprisingly, may labs appear to not be familiar with the partial reversibility of an aldehyde fixation.
METHODS:
Table 1 has been corrected according to the suggestion.
Line 217 has been corrected (our apology for the miss!)
FA inactivation: We thank for the suggestion and recommendation of different FA concentrations, but continue to stick with the standard dilutions as they are VERY commonly used in cell culture for fixing cultures and specimens. The text has been corrected and re-organized. Headlines (3.2.1 and 3.2.2) have been adjusted to clearly state the conditions and concentrations of FA in the various samples, and the text including the outcome was re-written to improve clarity.
TABLE G: We have re-worked this table: the three parallel blind passages are now indicated at the top ("A" to "C"); the negative control has been added for each virus (thank you very much for having pointed this important point out!).
RESULTS: (thank you for the suggestion!) We have now clearly marked that RT-stability (HIV), particle production (HIV) and viral infectivity (SARS) as different parameters can be assessed CONCLUSIONS: This section has been rephrased, leading to three bulleted statements as take-home messages. FIGURES/TABLES: All figures in question now contain a LOD for each virus, which correlate with the corresponding text. With these new corrections, text rephrasing/organization and additional information we hope to provide a legible and useful study that will be of interest to the audience of VIRUSES.
This manuscript is a resubmission of an earlier submission. The following is a list of the peer review reports and author responses from that submission.
Round 1
Reviewer 1 Report
Ulrike Seeburg, Thomas Klimkait and colleagues present an interesting study on the inactivation of mammalian viruses. The study requires several clarifications and additional quantifications. In addition, the text should be shortened and focused on the topic of the present study. Several problems are summarized in the following points; they greatly weaken the study:
1 The title of the article is not specific enough, and it is suggested to change it to a more directional title.
2 Each section of the results, including Figure 1, should have a running title and a conclusion.
3 The authors used multiple units of measurement in the manuscript to indicate the virus titer, such as moi, TCID50 and FFU, and the authors should refine the experimental methods.
4 The font in many Figures is too small and not clear enough, e.g. Figure 1, Figure 2, Figure 5. It will affect the understanding of the results and it is recommended to change it.
5 In Figure 3, the results of SARS-CoV-2 are suggested to be presented in the form of an ERROR BAR after the 3 duplicates are calculated.
6 In the experiment, for different viruses, the authors chose different buffer for virus inactivation and what was the basis of the choice.
7 Many English mistakes and inaccuracies should be corrected.
Some of the following mistakes appear several times in the manuscript, so please check and revise them.
Line 140 CO2 should be CO2.
Line 186 H2O should be H2O.
Line 252 106 should be 106.
Line 288,What does the means of “Treated wo lysis and Treated wo lysis, wo column D”? May be no lysis, no column ?
Line 335 1mL should be 1 mL.
Table 1 TCID50 should be TCID50.
Line 531 Postential should be Potential .
Line 547 The titers of AdV-5 and HIV-1 were 1x108 /mL and 1x105 /mL, Which method was used to determine the titer?
Reviewer 2 Report
Dear Authors,
Virus inactivation is a very important topic not only for experimenters but also for the people. The COVID crisis has highlighted the need to improve and clarify virus inactivation methods to avoid any contamination. The study is going in the right direction and even if the methodology and the experimental results are correct, there are major flows or forgotten that make the article irrelevant :
- The practical and regulatory bases for effective viral inactivation are not recalled even briefly (risk assessment, legal obligation, validation and effectiveness criteria)
- The choice of viruses and the titer assayed as well as the pathogen reduction were not discussed with regard to the surrogates described in the standards and the potentially infectious dose/risk assessment criteria
- The inactivation methods are too restrictive (what about heat treatment and detergents which are commonly used ) as well as the type of assayed buffers (composition, choice). In the same way, what about contaminated sera, blood samples or biopsies as only cell supernatant are described
- International norms or standards for assessing the virucidal activity are not described or mentioned
- Some graphs and figures are superfluous or poorly constructed (Tables E/1/2, Figure 4)
- Data concerning HIV-1 or SARS-CoV-2 stability in cell supernatant are interesting but off topic or irrelevant knowing that these viruses are manipulated in BSL-3 labs so high containment area. Perharps the same assay using clinical sample would make more sense as this kind of material is generally manipulated in low containment labs.
- Too much details are given at the expense of a clear message and a relevant introduction-conclusion-discussion
Regards